# Plasma Caffeine Levels and Risk of Alzheimer’s Disease and Parkinson’s Disease: Mendelian Randomization Study

**DOI:** 10.3390/nu14091697

**Published:** 2022-04-19

**Authors:** Susanna C. Larsson, Benjamin Woolf, Dipender Gill

**Affiliations:** 1Unit of Medical Epidemiology, Department of Surgical Sciences, Uppsala University, SE-751 85 Uppsala, Sweden; 2Unit of Cardiovascular and Nutritional Epidemiology, Institute of Environmental Medicine, Karolinska Institutet, SE-171 77 Stockholm, Sweden; 3MRC Integrative Epidemiology Unit, University of Bristol, Bristol BS8 2BN, UK; benjamin.woolf@bristol.ac.uk; 4School of Psychological Science, University of Bristol, Bristol BS8 1TU, UK; 5Faculty of Epidemiology and Population Health, London School of Hygiene and Tropical Medicine, London WC1E 7HT, UK; 6Department of Epidemiology and Biostatistics, School of Public Health, Imperial College London, London W2 1NY, UK; dipender.gill@imperial.ac.uk; 7Department of Clinical Pharmacology and Therapeutics, Institute for Infection and Immunity, St George’s, University of London, London SW17 0RE, UK; 8Genetics Department, Novo Nordisk Research Centre Oxford, Old Road Campus, Oxford OX3 7FZ, UK

**Keywords:** Alzheimer’s disease, caffeine, coffee, Mendelian randomization, Parkinson’s disease

## Abstract

We leveraged genetic variants associated with caffeine metabolism in the two-sample Mendelian randomization framework to investigate the effect of plasma caffeine levels on the risk of Alzheimer’s disease and Parkinson’s disease. Genetic association estimates for the outcomes were obtained from the International Genomics of Alzheimer’s Project, the International Parkinson’s Disease Genomics consortium, the FinnGen consortium, and the UK Biobank. Genetically predicted higher plasma caffeine levels were associated with a non-significant lower risk of Alzheimer’s disease (odds ratio 0.87; 95% confidence interval 0.76, 1.00; *p* = 0.056). A suggestive association was observed for genetically predicted higher plasma caffeine levels and lower risk of Parkinson’s disease in the FinnGen consortium. but not in the International Parkinson’s Disease Genomics consortium; no overall association was found (odds ratio 0.92; 95% confidence interval 0.77, 1.10; *p* = 0.347). This study found possible suggestive evidence of a protective role of caffeine in Alzheimer’s disease. The association between caffeine and Parkinson’s disease requires further study.

## 1. Introduction

Caffeine is the most commonly consumed psychostimulant [1]. It readily crosses the blood–brain barrier and exerts most of its biological effects via adenosine receptor antagonism, thereby affecting basic human processes, such as sleep, arousal, cognition, and memory [1]. Experimental data suggest that caffeine has neuroprotective effects on various neurodegenerative diseases via several mechanisms, including reduction in excitatory neurotransmitter release and neuroinflammatory responses [1]. Hence, it has been postulated that caffeine may play a role in Alzheimer’s disease and Parkinson’s disease. However, epidemiological studies investigating these associations are scarce, and studies on the consumption of coffee, the main source of caffeine in many populations, in relation to these diseases have yielded inconclusive findings [2,3]. This inconsistency may be related to methodological issues, such as residual confounding (e.g., from smoking, which is related to both coffee intake and Parkinson’s disease) or misclassification of caffeine exposure. Furthermore, other substances in roasted coffee, such as chlorogenic acids and melanoidins, may affect the risk of neurodegenerative diseases, so that any effect of caffeine per se is not clear.

Caffeine is primarily metabolized in the liver via the cytochrome P450 isoform CYP1A2 [4]. The level of CYP1A2 is controlled by the *AHR* gene. Single-nucleotide polymorphisms (SNPs) near the genes encoding CYP1A2 and AHR have been shown to be associated with plasma caffeine levels [4]. Here, we leveraged these variants in the two-sample Mendelian randomization (MR) framework to investigate the causal effect of plasma caffeine levels on Alzheimer’s disease and Parkinson’s disease risk.

## 2. Materials and Methods

From the largest published meta-analysis of genome-wide association studies (GWAS) of plasma caffeine levels (*n* = 9876 individuals of European ancestry), we selected the two SNPs with the strongest association with plasma caffeine located near *CYP1A2* (rs2472297; *p* = 1.0 × 10^−20^) and *AHR* (rs4410790; *p* = 1.8 × 10^−13^) respectively [4]. The SNP-outcome association estimates for the caffeine-associated variants were obtained from the International Genomics of Alzheimer’s Project consortium [5], the International Parkinson’s Disease Genomics consortium (excluding 23andMe) [6], the FinnGen consortium (freeze 6) [7] and the UK Biobank study. For the consortia, we used publicly available summary-level data. For UK Biobank, we performed a GWAS. After excluding UK Biobank participants who had withdrawn consent, we defined cases as any participants who were algorithmically classified as having Alzheimer’s disease (UKB ID 42021, *n* = 954) and non-cases as any participants who were not (*n* = 487,331). The analysis was conducted using BOLT-LLM, adjusted for age, sex, genotyping chip, and the first 10 genetic principal components, in the Medical Research Council-Integrative Epidemiology Unit UK Biobank GWAS pipeline. Full methods have been described elsewhere [8]. Numbers of cases and non-cases in each study are given in Figure 1.

The fixed-effects inverse-variance weighted method was used for MR analysis. Individual MR estimates for each SNP were generated by dividing the SNP-outcome association by the SNP-exposure association. Estimates across studies for each outcome were combined through the fixed-effects meta-analysis, and heterogeneity between estimates was computed using the Cochran’s *Q* test. The analyses were performed in Stata (StataCorp, College Station, TX, USA). All reported *p*-values are two-tailed.

## 3. Results

Genetically predicted higher plasma caffeine levels were associated with a non-significant lower risk of Alzheimer’s disease in the meta-analysis of the three studies (odds ratio 0.87; 95% confidence interval 0.76, 1.00; *p* = 0.056) (Figure 1).

There was a suggestive association between genetically predicted higher plasma caffeine levels and reduced risk of Parkinson’s disease in the FinnGen consortium but not in the International Parkinson’s Disease Genomics consortium; no overall association was found (odds ratio 0.92; 95% confidence interval 0.77, 1.10; *p* = 0.347) (Figure 1).

## 4. Discussion

The suggestive finding for genetically predicted higher plasma caffeine and reduced risk of Alzheimer’s disease is consistent with experimental data, which indicate that caffeine may have a protective role [1]. Furthermore, a case-control study found that plasma caffeine levels were associated with reduced odds of dementia or delayed onset of dementia, particularly in individuals with mild cognitive impairment [9]. The association with consumption of coffee, which is the main source of caffeine in many populations, has not been strongly associated with risk of Alzheimer’s disease or dementia in the few cohort studies assessing this association [2]. Nevertheless, a recent study showed that high coffee consumption was associated with slower cognitive decline and lower likelihood of transitioning to mild cognitive impairment or Alzheimer’s disease, as well as slower Aβ-amyloid accumulation [10]. Another study in 411 healthy older adults demonstrated that higher lifetime levels of coffee consumption might contribute to lowering the risk of Alzheimer’s disease or related cognitive decline by reducing pathological cerebral amyloid deposition [11]. Moreover, green and roasted coffee extracts and their main components (i.e., 5-O-caffeoylquinic acid and melanoidins) have been shown to hinder Aβ on-pathway aggregation and toxicity in a human neuroblastoma SH-SY5Y cell line [12]. Caffeine may affect arginine metabolism, and a recent study showed that arginase inhibition reverses cognitive decline in Alzheimer’s disease mice [13]. Several studies have linked the upregulation of arginase to a variety of other central nervous system diseases, including Parkinson’s disease [14]. A recent study investigating comorbid Alzheimer’s disease found that Alzheimer’s disease is strongly associated with Parkinson’s disease [15].

Coffee consumption has been reported to be inversely associated with Parkinson’s disease risk in several, but not all, epidemiological studies exploring this [3]. Smoking appears to protect against Parkinson’s disease, and could bias the results for coffee consumption and Parkinson’s disease in observational studies if adjustment for smoking is incomplete, e.g., due to misclassification of smoking history and intensity. While this MR study found a suggestive inverse association between genetically predicted plasma caffeine levels and Parkinson’s disease in FinnGen, this association was not supported in the International Parkinson’s Disease Genomics consortium. Possible reasons for these disparate findings could be a chance finding in FinnGen, or differences in the study populations. For example, Finland has the highest per capita consumption of coffee in the world, and the potentially higher caffeine exposure in the FinnGen study might explain why an association was only observed in this study.

Previously published MR studies on genetically predicted coffee consumption and risk of Alzheimer’s disease [16,17] and all-cause dementia [18] have found trends in the opposite direction to our findings with regard to genetically predicted plasma caffeine levels in relation to Alzheimer’s disease. This is as expected, because the considered genetic variants associated with slower caffeine metabolism and thus higher plasma caffeine levels are also related to lower coffee consumption [19]. This also makes physiological sense, as those individuals who metabolize caffeine more slowly will tend to drink less coffee to achieve the same levels of plasma caffeine. A previous MR study found no association between genetically predicted coffee consumption and Parkinson’s disease [20], which is in line with our overall null results for genetically predicted plasma caffeine levels and this disease.

It should be noted that the genetic instrument used in our MR study aimed to capture caffeine exposure rather than coffee consumption per se. Thus, our findings for genetically predicted caffeine exposure are not directly comparable to the possible effects of coffee drinking, as coffee contains other bioactive compounds that may prevent amyloid aggregation [12] and potentially reduce the risk of Alzheimer’s disease.

A strength of our study is the MR design. MR studies that leverage genetic variants in genomic regions (e.g., *CYP1A2*) with a biological link to the metabolism of the exposure of interest (e.g., caffeine) are less likely to be affected by confounding compared to traditional observational studies. Additionally, we analyzed the association between genetically predicted plasma caffeine and Alzheimer’s disease in two independent studies, and the association was consistent in both studies, albeit with non-significant results, possibly due to low power. The observed consistency strengthens the evidence for causality. A potential shortcoming of our work is that the results might only be generalizable to European populations. In addition, with only two genetic instruments and summary-level data, we could not use statistical methods to investigate possible pleiotropy biasing the MR estimates, or non-linear relationships between plasma caffeine and disease risk. We therefore cannot rule out a threshold effect, with a protective effect only seen, for example, with very high plasma levels of caffeine.

## 5. Conclusions

This MR study showed that genetically predicted higher plasma caffeine levels were associated with a non-significant lower risk of Alzheimer’s disease. The potential role of caffeine in Alzheimer’s and Parkinson’s disease needs further research.

## Figures and Tables

**Figure 1 nutrients-14-01697-f001:**
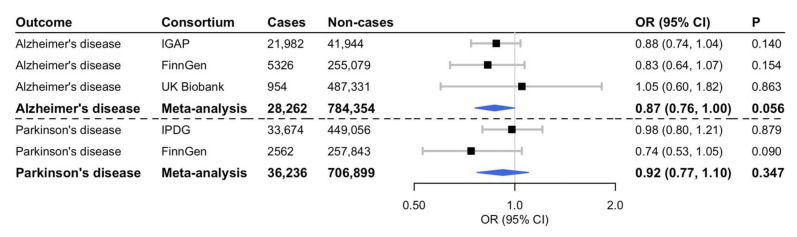
Associations between genetically predicted higher plasma caffeine levels and risk of Alzheimer’s and Parkinson’s disease. CI, confidence interval; OR, odds ratio. Cochran’s *Q* test statistic for heterogeneity between study-specific estimates: *Q* = 0.59 (*p* = 0.74) for Alzheimer’s disease and *Q* = 1.73 (*p* = 0.19) for Parkinson’s disease. The blue diamonds represent the combined OR estimate with its 95% CI from the meta-analysis of all studies for each outcome.

## Data Availability

Data generated and analyzed during the study are available from the corresponding author by request.

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
