# Peer review of "Plasma Caffeine Levels and Risk of Alzheimer’s Disease and Parkinson’s Disease: Mendelian Randomization Study"

_nutrients, 2022, doi:10.3390/nu14091697_

Round 1

Reviewer 1 Report

Summary: In the manuscript entitled "Plasma Caffeine Levels and Risk of Alzheimer's Disease and Parkinson's Disease: Mendelian Randomization Study," Susanna C. Larsson et al. leveraged genetic variants associated with caffeine metabolism in the two-sample Mendelian randomization framework to investigate the effect of plasma caffeine levels on risk of Alzheimer's disease and Parkinson's disease.

Overall, this is potentially a good report presenting several sets of novel and highly relevant results demonstrating a suggestive association between genetically-predicted higher plasma caffeine levels and lower risk of Parkinson's disease. Moreover, this study found possible suggestive evidence of a protective role of caffeine in Alzheimer's disease. However, in its current form, this study has several minor issues that should be carefully addressed.

Critiques:

General:

The introduction section is too short. A more comprehensive literature analysis with recent articles included may improve the quality of the paper.

  1. For instance, a study by Kim et al. (2019) demonstrates that higher lifetime coffee intake contributes to lowering the risk of AD or related cognitive decline by reducing pathological cerebral amyloid deposition (1).
  2. Another very recent study by Samantha L. Gardener et al. (2021) proves that higher baseline coffee consumption is associated with slower cognitive decline and a lower likelihood of transitioning to mild cognitive impairment or AD status. Higher baseline coffee consumption is also associated with slower Aβ-amyloid accumulation (2).
  3. Additionally, a recent paper by Avitan et al. (2021) investigates the AD comorbidity and shows that Alzheimer's is strongly associated with Parkinson's disease (3), which is an interesting observation in the context of the present study.

The methods section can be improved as well with a more detailed description of the method and statistical analysis approach. Referencing several analogous studies might be helpful.

The figure is of not sufficient quality. The odds ratio plot could be done in different colors to improve its visibility and underline the associations.

The authors could speculate about the possible mechanisms of the correlation between plasma caffeine levels and the risk of Alzheimer's and Parkinson's Diseases. I suggest one possible mechanism…

  1. More recently, Polis et al. (2018) demonstrated that arginase inhibition reverses cognitive decline in Alzheimer's disease mice (4).
  2. Remarkably, several studies linked the upregulation of arginase to a variety of other CNS diseases, including Parkinson's disease (5).

References:

  1. Kim, J.W., Byun, M.S., Yi, D. et al. Coffee intake and decreased amyloid pathology in human brain. Transl Psychiatry 9, 270 (2019). https://doi.org/10.1038/s41398-019-0604-5
  2. Gardener SL, Rainey-Smith SR, Villemagne VL, Fripp J, Doré V, Bourgeat P, Taddei K, Fowler C, Masters CL, Maruff P, Rowe CC, Ames D, Martins RN; AIBL Investigators. Higher Coffee Consumption Is Associated With Slower Cognitive Decline and Less Cerebral Aβ-Amyloid Accumulation Over 126 Months: Data From the Australian Imaging, Biomarkers, and Lifestyle Study. Front Aging Neurosci. 2021 Nov 19;13:744872. doi: 10.3389/fnagi.2021.744872.
  3. Avitan I, Halperin Y, Saha T, Bloch N, Atrahimovich D, Polis B, Samson AO, Braitbard O. Towards a Consensus on Alzheimer's Disease Comorbidity? J Clin Med. 2021 Sep 24;10(19):4360. doi: 10.3390/jcm10194360. PMID: 34640387; PMCID: PMC8509357.
  4. Polis B, Srikanth KD, Elliott E, Gil-Henn H, Samson AO. L-Norvaline Reverses Cognitive Decline and Synaptic Loss in a Murine Model of Alzheimer's Disease. Neurotherapeutics. 2018 Oct;15(4):1036-1054. doi: 10.1007/s13311-018-0669-5. PMID: 30288668; PMCID: PMC6277292.
  5. Caldwell, R. W., Rodriguez, P. C., Toque, H. A., Narayanan, S. P., & Caldwell, R. B. (2018). Arginase: A Multifaceted Enzyme Important in Health and Disease. Physiological reviews, 98(2), 641–665. https://doi.org/10.1152/physrev.00037.2016

Reviewer 2 Report

I read with interest the Communication by Larsson et al. The topic of issue is of interest although highly debated. The manuscript is well written. Recently, some interesting papers published contrasting results (Zhang et al. J Alzheimers Dis. 2021 doi: 10.3233/JAD-210678, Zheng & Niu Frontiers in Nutrition 2022 doi: 10.3389/fnu.2022.850004) point out in a different direction as compared to the AA’s main results of a possible association between plasma caffeine levels and decreased Alzheimer’s disease (AD) risk. In addition, the p-value (i.e., 0.056) (two tailed ? or one tailed ?) is just above the standard 0.05 statistical significant threshold usually considered in biomedical field. As it concerns the possible plasma caffeine levels-Parkinson’ disease (PD) relationship, the meta-analysis data results in a p-value of 0.347, once again above the usually accepted statistically significant threshold. In addition, other potential confounding factors should be considered beyond the ones reported for the Authors’ BOLT-LLM analysis (i.e., line 65 “age, sex, genotyping chip, and the first 10 genetic principal compoenents”). The overall meta analysis data still inconclusive and certainly needing further research, exactly as the AA conclude in their final paragraph.

Reviewer 3 Report

This communication, which was carried out by analyzing different cohorts of patients and non-pathological subjects, could benefit from some experimental data that the authors could evaluate.

A possible experimental reason, which supports the authors' conclusion, could be the following indication:

..."The considerable caffeine content of coffee (especially roasted coffee) might be another limitation to its use for prophylactic purposes; nevertheless, here we showed that caffeine does not hinder Aβ aggregation and neurotoxicity and is therefore not required for the coffee extract’s neuroprotective activity, suggesting that decaffeinated coffee could be used for this purpose"...  (ref. https://doi.org/10.1016/j.foodchem.2018.01.075 )

The authors could better support their thesis adding these results in discussion section (lane 90).

Since the coffe extracts  (chlorogenic acids and melanoidins but not caffeine) seem to prevent amyloid aggregation, individuals with mild cognitive impairment might have a better benefit. What is the cognitive impairment of AD patients who have been  evaluated in the different biobanks? (In fig 1, you could complete the characteristics of investigated subjects)

AHR (lane 48): it is not described in full

Round 2

Reviewer 2 Report

The AA have properly addressed my concerns and criticism. The Ms content has substantially benefited from the revision process. Although I do feel that the issue deserves further study from both epidemiologic standpoint and a better in depth biochemical understanding, I personally find no obstacles in accepting the revised Ms.